# Mode of Death after Extracorporeal Cardiopulmonary Resuscitation

**DOI:** 10.3390/membranes11040270

**Published:** 2021-04-08

**Authors:** Viviane Zotzmann, Corinna N. Lang, Xavier Bemtgen, Markus Jäckel, Annabelle Fluegler, Jonathan Rilinger, Christoph Benk, Christoph Bode, Alexander Supady, Tobias Wengenmayer, Dawid L. Staudacher

**Affiliations:** 1Department of Cardiology and Angiology I (Heart Center Freiburg—Bad Krozingen), Medical Center-University of Freiburg, Faculty of Medicine, University of Freiburg, 79106 Freiburg im Breisgau, Germany; corinna.nadine.lang@uniklinik-freiburg.de (C.N.L.); xavier.bemtgen@uniklinik-freiburg.de (X.B.); markus.jaeckel@uniklinik-freiburg.de (M.J.); annabelle.fluegler@uniklinik-freiburg.de (A.F.); jonathan.rilinger@uniklinik-freiburg.de (J.R.); christoph.bode@uniklinik-freiburg.de (C.B.); alexander.supady@uniklinik-freiburg.de (A.S.); tobias.wengenmayer@uniklinik-freiburg.de (T.W.); dawid.staudacher@uniklinik-freiburg.de (D.L.S.); 2Department of Medicine III (Interdisciplinary Medical Intensive Care), Medical Center-University of Freiburg, Faculty of Medicine, University of Freiburg, 79106 Freiburg im Breisgau, Germany; 3Department of Cardiovascular Surgery (University Heart Center Freiburg—Bad Krozingen), Medical Center-University of Freiburg, Faculty of Medicine, University of Freiburg, 79106 Freiburg im Breisgau, Germany; christoph.benk@uniklinik-freiburg.de; 4Institute of Global Health, University of Heidelberg, 69120 Heidelberg, Germany

**Keywords:** ECPR, extracorporeal cardiopulmonary resuscitation, CPR, resuscitation, mode of death, withdrawal from therapy, hemodynamic shock, brain death

## Abstract

Introduction: Extracorporeal cardiopulmonary resuscitation (ECPR) might be a lifesaving therapy for patients with cardiac arrest and no return of spontaneous circulation during advanced life support. However, even with ECPR, mortality of these severely sick patients is high. Little is known on the exact mode of death in these patients. Methods: Retrospective registry analysis of all consecutive patients undergoing ECPR between May 2011 and May 2020 at a single center. Mode of death was judged by two researchers. Results: A total of 274 ECPR cases were included (age 60.0 years, 47.1% shockable initial rhythm, median time-to-extracorporeal membrane oxygenation (ECMO) 53.8min, hospital survival 25.9%). The 71 survivors had shorter time-to-ECMO durations (46.0 ± 27.9 vs. 56.6 ± 28.8min, p < 0.01), lower initial lactate levels (7.9 ± 4.5 vs. 11.6 ± 8.4 mg/dL, p < 0.01), higher PREDICT-6h (41.7 ± 17.0% vs. 25.3 ± 19.0%, p < 0.01), and SAVE (0.4 ± 4.8 vs. −0.8 ± 4.4, p < 0.01) scores. Most common mode of death in 203 deceased patients was therapy resistant shock in 105/203 (51.7%) and anoxic brain injury in 69/203 (34.0%). Comparing patients deceased with shock to those with cerebral damage, patients with shock were significantly older (63.2 ± 11.5 vs. 54.3 ± 16.5 years, p < 0.01), more frequently resuscitated in-hospital (64.4% vs. 29.9%, p < 0.01) and had shorter time-to-ECMO durations (52.3 ± 26.8 vs. 69.3 ± 29.1min p < 0.01). Conclusions: Most patients after ECPR decease due to refractory shock. Older patients with in-hospital cardiac arrest might be prone to development of refractory shock. Only a minority die from cerebral damage. Research should focus on preventing post-CPR shock and treating the shock in these patients.

## 1. Introduction

The survival to hospital discharge after an out-of hospital (OHCA) and in-hospital cardiac arrest (IHCA) is 10% and 15%, respectively [1].

A considerable number of patients after cardiac arrest never regain cardiac function and die on scene. Even when return of spontaneous circulation (ROSC) can be achieved, outcome is dire [2]. In patients with a sustainable ROSC, a complex pathophysiological process commonly occurs, called post-cardiac arrest syndrome [3]. During their subsequent in-hospital course, many of these patients will die and data suggests up to 60–70% in hospital mortality in both IHCA and OHCA [3,4,5]. The pathophysiology of the post-resuscitation shock usually combines vasoplegia and myocardial dysfunction, resulting in refractory circulatory failure and multiple organ dysfunctions [2]. Apart from shock, anoxic-ischemic brain damage is a major driver of adverse outcome after cardiopulmonary resuscitation (CPR).

In OHCA, recent data suggests that patients frequently die due to neurological damage [6], which is sustained during the anoxic, no-flow period of cardiac arrest or as a result of reperfusion injury that occurs in the early post-resuscitation phase [6]. In IHCA, patients seem more likely to present with pre-existing comorbidities, which is a strong determinant for withdrawal of life sustaining therapies in these patients [6].

However, often data on outcome after CPR only report a composite endpoint of death. Information on and categorization of the cause of death could provide valuable information to assess benefits of targeted interventions or quality-improvement initiatives, which might guide the development of new treatments and improvement of outcomes. Overall, an adequate appraisal of causes of death is a cornerstone in the management of the post-resuscitation patient in order to identify potential therapeutic tools. A categorization for the mode of death after cardiac arrest in both IHCA and OHCA has been proposed by Witten and coworkers [6].

However, it is not clear whether this data can be applied to patients who underwent extracorporeal cardiopulmonary resuscitation (ECPR). Due to the longer low-flow time, a higher percentage of neurological damage would be conceivable. We therefore conducted a retrospective registry analysis in order to categorize the mode of death in ECPR.

## 2. Methods

### 2.1. Study Setting

This study includes all patients who were admitted between May 2011 and May 2020 to the medical intensive care unit at our university hospital and received an extracorporeal cardiopulmonary resuscitation (ECPR). Patients who received ECPR in the emergency room and were then transferred to a surgical department were excluded. Furthermore, patients were excluded if the therapy had already been stopped before reaching the intensive care unit.

ECPR was defined as veno-arterial extracorporeal membrane oxygenation (VA-ECMO) implantation during continuous CPR without ROSC or as VA-ECMO implantation within the first 20 min. after ROSC with uncontrollable hemodynamic instability [7]. All VA-ECMO cases treated at our institution were detected by a computerized search for the German OPS-codes (operation- and procedure-keys) for VA-ECMO (8-852.30–8-852.30e), followed by manual review on case-by-case basis. All patients treated at the interdisciplinary medical intensive care unit were considered for this retrospective registry analysis, excluding patients cannulated for VA-ECMO in the operation theatre and treated at a surgical intensive care unit.

### 2.2. Local ECPR Algorithm

Starting in 2011, all patients with IHCA without ROSC after 15 min are routinely screened for an indication for ECPR. As defined by our standard operating procedures, unwitnessed cardiac arrest, prolonged duration of CPR without signs of life (breathing, swallowing etc.), a non-shockable initial rhythm, life-threatening bleeding and advanced age ≥75 years are considered relative contraindications for ECMO cannulation [8]. Final decision to cannulate is driven by a team decision at the bedside including at least two physicians, a perfusionist, and two nurses. Local standard operating procedure suggests cannulation for ECPR after IHCA on-site or in the cardiac catheterization laboratory, whichever results in shorter low-flow durations. For OHCA, emergency medical services personnel are encouraged to transport patients without ROSC with ongoing mechanical chest compressions to our center, where the patients are screened for ECPR. The same contraindications apply as for IHCA patients. OHCA patients were either routed to the emergency room or the cardiac catheterization laboratory, according to the presumed cause for collapse, non-cardiac or cardiac, respectively [9]. Screening for ECPR and cannulation were then performed in the emergency room or the catheterization laboratory, whichever destination had been chosen. Starting in August 2018, an out-of-hospital ECPR service was implemented and offers on-site out-of-hospital ECPR during working hours.

### 2.3. ECMO Cannulation and Maintenance

Local standard operating procedures suggest cannulation for ECPR to be performed in Seldinger’s technique by two experienced intensivists and one perfusionist. SCPC (Sorin Centrifugal Pump Console, LivaNova, London, United Kingdom) or Cardiohelp (Maquet Getinge Group, Rastatt, Germany) systems could be used. Typical venous (draining) cannulas were 21–23 Fr (French—Charrière) in diameter and 55 cm of length while arterial (returning) cannulas were 15–17 Fr, 23 cm (both HLS cannula, Maquet Getinge Group, Rastatt, Germany). For patients without life-threatening bleeding, anticoagulation was provided by intravenous administration of unfractionated heparin aiming at a partial thromboplastin time of 50–60 sec. The management of vasopressors and fluid therapy was driven by clinical judgement of the ECMO-experienced intensivist in charge and has been reported earlier [10,11].

### 2.4. Coronary Angiography after ECPR

Early detection and treatment of the cause for collapse was attempted in all ECPR cases. A coronary angiography was therefore performed in all patients with a presumed cardiac cause. In case of presumed non-cardiac cause and non-conclusive findings in ultrasound and computed tomography studies, a coronary angiography is advertised by local standard operating procedures. Intervention of a coronary stenosis detected by angiography was performed if recommended by current guidelines [12].

### 2.5. Neuroprognostication

Withdrawal of therapy due to poor neurological outcome requires advanced neuroprognostication to assess neurological outcomes. In our center, this consists of multimodal diagnostic (in accordance with the guidelines of the ESC 2015 [3] and the German Society for Neurology 2018) including an interdisciplinary discussion. Reliable neuro-prognostication is carried out at the earliest after 72 h and at least 24 h after reaching normothermia. Within 72 h it is carried out only if there are signs of brain death or if there is evidence of brain herniation. Specifically, neuroprognostication is performed following a local standard operating procedure including a clinical neurological examination (RASS, brainstem reflexes, myoclonus, epileptic seizures), an electrophysiological examination (Medianus-SEP, EEG), the level and dynamics of the biomarkers in the serum (NSE, S-100 [13]), and cerebral imaging.

### 2.6. Definition of Refractory Shock during VA-ECMO

Cardiogenic shock describes a condition with inadequate end-organ perfusion due to reduced cardiac output, characterized by low blood pressure, volume overload, and end-organ hypoperfusion.

Initially, the disruption of the macrocirculation due to a low-output syndrome or even a complete lack of cardiac ejection is observed. With VA-ECMO or ECPR, in principle, the macrocirculation can be restored and stabilized. However, a pronounced post-resuscitation syndrome often leads to a therapy-refractory shock. An excessive inflammatory reaction (SIRS) leads to an increased leakage with increasing vasopressor- and albumin-requirements. The microcirculation, which is increasingly disturbed by vasopressors, then leads to a worsening of hypotension due to an inflammatory mediator cascade including NO and peroxynitrite causing cardiodepression, capillary leakage, and vasoplegia. This is accompanied with increasing lactate levels. In patients with severe shock including extravascular volume shift and vasoplegia, stabilization cannot be achieved even by VA-ECMO. As for this research, patients were considered dying due to refractory shock when under full VA-ECMO support and high-dose vasopressor therapy, repeated high volume and blood product-administration, yet failing blood pressure targets (mean arterial pressure <65 mmHg).

### 2.7. Data Collection

Presented data derive from a single-center retrospective registry analysis and were blinded to patient identity and covered by an ethics approval (Ethics Committee of Albert-Ludwigs University Freiburg, file number 151/14 and 533/19).

Cause of death was defined following the publication of Witten et al. [6] which classified the mode of death in five categories. In order to adapt the categories to patients after ECPR, we modified the categorization and deleted the category “sudden cardiac death”, which cannot occur on ongoing VA-ECMO and added “suspected patient will” as a 5th category (see Table 1).

This category includes the withdrawal of further treatment if the patient’s presumed will was not to be resuscitated, or withdrawal of therapy and termination of intensive care treatment due to an expected poor quality of life (e.g., in context of previously existing serious illness such as dementia or an advanced cancer disease).

The next category “Persisting cardiogenic or direct post-resuscitation shock” includes withdrawal of therapy in either progressive, refractory hemodynamic shock due to refractory vasoplegia in post-resuscitation shock, with inadequate VA-ECMO despite aggressive catecholamine therapy and volume substitution. Or, withdrawal in case of lack of hemodynamic stabilization with persisting dependency of a cardiac support system (VA-ECMO or Impella^®^) without the possibility of definitive care using an LV-assist device (LVAD) or heart transplantation.

Persistent liver failure, as well as multi-organ failure or an uncontrollable septic shock are grouped into the next category “multi-organ failure”.

For this research, mode of death in ECPR was judged by two intensivists independently according to the medical electronic files and assigned to one of the five categories for each individual patient. If the two investigators came to different judgements, the case was discussed, and a consensus was found on the most likely mode of death. In addition to the mode of death, we recorded the patient characteristics, the data and temporal course of the resuscitation and implantation of the ECPR, the brain necrosis markers and parameters that reflect the severity of the disease (for e.g., lactate etc.).

### 2.8. Statistical Analysis

For data analysis, SPSS (version 23, IBM) or Prism (version 9, GraphPad) were used and a p-value of <0.05 was considered statistically significant. All data are given as [mean ± standard deviation] if not stated otherwise. Unpaired t-test (if Gaussian distribution was assumed as tested by Kolmogorov-Smirnov normality test) or Mann–Whitney test (in cases where normal distribution could not be assumed) were used. Fisher’s exact test was used to interpret contingency tables and Chi^2^. One-way ANOVA or Chi^2^-test were used to compare continuous or discontinuous variables in three groups, respectively.

## 3. Results

A total of 274 patients after ECPR could be included. Patients were mostly male, mean age was 60 years (range 18 to 87 years) and hospital survival was 25.9%. All patients’ characteristics are given in Table 2. Outcome data and clinical parameters during ICU stay are given in Table 3.

Comparing survivors to non-survivors we found that survivors had significantly lower low-flow durations, lower initial lactate levels, a lower PREDICT score, and more often presented with a shockable initial rhythm. Due to (massive) bleeding complications in a quarter of the cases, the target temperature was set to 36 °C degrees in, while targeted temperature management (TTM) with 33 °C was used in 75.9% of the cases.

A total of 203 patients died during the hospital stay. Many patients died early after cannulation for VA-ECMO, specifically 95/203 (46.8%) within the first two and 133/203 (65.5%) within first four days.

When grouping patients according to the mode of death, the most dominant reason for death was therapy resistant shock despite running VA-ECMO in 105/203 (51.7%) of cases. The second most important reason for death was severe cerebral damage in 69/203 (34.0%) of cases, see Figure 1 and Table 2. 

When focusing on the timeline of death, shock as well as cerebral damage predominantly complicated early the phase of therapy after VA-ECMO cannulation, see Figure 2 and Figure 3.

When comparing characteristics of patients with a putative mode of death by shock to those with cerebral damage, we found that these patient groups were different in several key characteristics as given in Table 4. Specifically, patients with death following cerebral damage were significantly younger than those who die by shock, were significantly more often resuscitated out of hospital, and had a significantly longer low flow duration. Moreover, NSE and S100 values of day 1 were significantly higher in case of neurological death, see Table 4 and Figure 4A,B.

## 4. Discussion

We found an overall hospital survival of 25.9% in a large registry of extracorporeal cardiopulmonary resuscitation. The observed overall hospital survival rate in our sample is well in line with reports from the ELSO international ECLS registry with reported survival of 29% in adult ECPR [14] or with large European registries with a hospital survival of 23% [15].

After ECPR, the two most prominent causes of death were persistant shock despite VA-ECMO therapy followed by cerebral damage. This is a surprising finding since VA-ECMO is considered an effective treatment of various different subtypes of shock including cardiogenic shock (genuine [16] or sepsis induced [17]), obstructive shock [18] or even septic shock [19]. It is known that patients with VA-ECMO require significant doses of vasopressors during extracorporeal support [20], which might indicate a persistent shock despite VA-ECMO in a considerable number of patients. Similar results have been reported for lactate as a surrogate for tissue hypoperfusion, which remains high in some patients despite VA-ECMO [21]. The frequent development of end-organ failure including acute kidney injury in patients on VA-ECMO [22,23], further suggests insufficient end-organ perfusion in some patients. We show that, compared to patients with cerebral damage, patients with persistent shock were older, more frequently underwent in-hospital cardiac arrest, and had shorter low-flow durations. Further research is needed in order to understand the reasons for persistent shock despite VA-ECMO, which might help improving survival in this group.

The second most common cause of death was cerebral damage. It has been reported earlier that serum markers of neuronal damage like NSE are elevated after ECPR and correlate with survival [24]. Since neuroprognostication cannot be based on serum markers alone [25] current guidelines recommend delaying multimodal prognostication to day 3 after CPR [26] in patients after conventional CPR. Despite this recommendation, which was incorporated in our local standard operating procedures, a considerable amount of patients in our registry died due to cerebral damage before day 3. Considering the long conventional CPR-durations in ECPR patients [15], a high incidence of devastating brain damage in ECPR has to be presumed [27] and has been demonstrated earlier [28]. In these patients, a neuroprognostication before day 3 is reasonable.

When comparing patients dying of shock to those dying of cerebral damage, we found that patients dying of shock were older, had lower low flow durations, and lower NSE values. When judicating the mode of death, also NSE values might have been factored in; therefore, NSE might have been a confounder. As for age, incidence of septic shock is significantly increased in older patients [29]. One might speculate that older patients are more prone to shock. As for low flow duration, it is known that cardiac output is significantly reduced during CPR which causes ischemia in the whole body, first and foremost in the brain. Longer low flow durations therefore cause more brain ischemia leading to more brain death [27].

## 5. Limitations

This is a retrospective observational study and therefore contains the risk of selection and reporting bias. Moreover, this is a single-center report and specific processes may influence the presented results. In order to reduce bias, hard endpoints like hospital mortality were chosen. Data was checked for consistency by two researchers independently (especially focusing on inclusion of only ECPR cases). Mode of death was also judged twice and in case of divergent judgements cases were openly discussed and a team decision was made. Moreover, a percentage of patients who died early due to refractory shock might have died later due to other causes like brain damage, which might result in an underreporting of these causes of death.

Regarding the significantly increased nicotine rate among the survivors, we think that there is a bias here. For those who survived, the personal anamnesis could probably be supplemented better afterwards, than in the cases in which we had to rely on third-party anamnesis.

## 6. Conclusions

The most common reason for death in patients after ECPR is therapy refractory cardiogenic shock. Older patients with in-hospital cardiac arrest might be prone to the development of shock. Research should focus on preventing post-CPR shock and treating the shock in these patients.

## Figures and Tables

**Figure 1 membranes-11-00270-f001:**
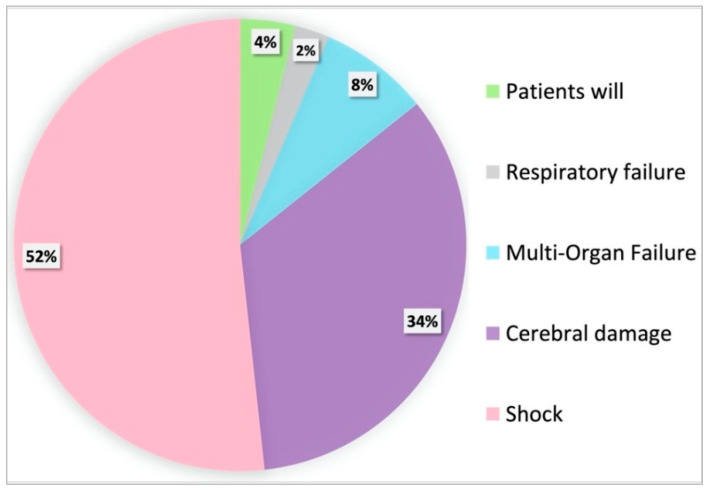
Proportions of the different modes of death given as a percentage of all non-survivors. The most prominent causes of in-hospital death after ECPR were refractory shock followed by cerebral damage.

**Figure 2 membranes-11-00270-f002:**
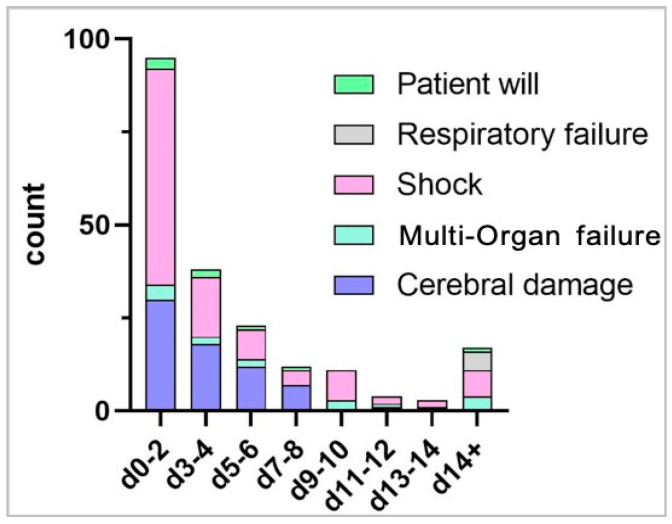
Timeline of Mode of Death after ECPR. A substantial number of patients died within the first two days after VA-ECMO cannulation. While death due to cerebral damage was prominent within the first week, shock was the major driver through the whole clinical course.

**Figure 3 membranes-11-00270-f003:**
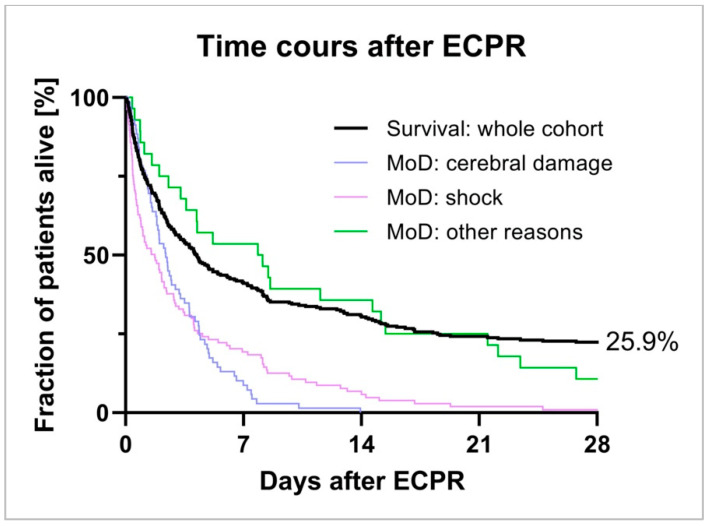
Time course after ECPR. Death after ECPR in the whole cohort and in selected subgroups.

**Figure 4 membranes-11-00270-f004:**
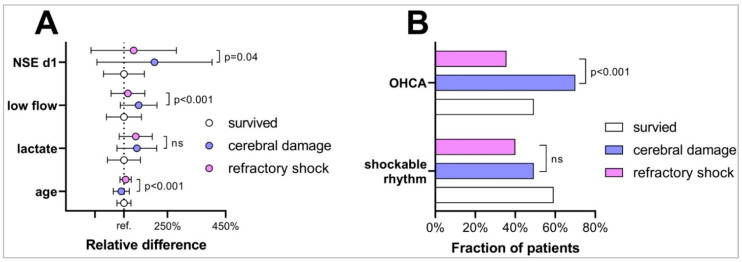
Characteristics of ECPR patients in respect to outcome. (**A**) Relative difference of characteristics of patients dying after ECPR due to cerebral damage (blue) or refractory shock (pink) compared to ECPR survivors (white, reference). Those dying from shock were older, had lower low flow durations, and lower NSE values on day. (**B**) Fraction of patients presenting after OHCA and shockable rhythm comparing survivors deceased patients due to cerebral damage (blue) or refractory shock (pink).

**Table 1 membranes-11-00270-t001:** Categorization of the mode of death in extracorporeal cardiopulmonary resuscitation (ECPR) patients. Adaptation of reasons for death following ECPR of the categories adopted from Witten et al. [6].

Reasons for Death Following ECPR in Five Categories	
Neurological Withdrawal	Withdrawal of care based on expectations of a poor neurological recovery. If an assessment off sedation is not done, e.g., in the early hours during targeted temperature management (TTM), there must be other evidence of severe neurologic injury (e.g., severe cerebral edema or herniation).
Persisting Cardiogenic/Post-Resuscitation Shock	Withdrawal from therapy in either progressive, refractory hemodynamic shock due to refractory vasoplegia in post-resuscitation shock, with inadequate VA-ECMO despite aggressive catecholamine therapy and volume substitution. Or, Withdrawal in case of lack of hemodynamic stabilization with persisting dependency of a cardiac support system (VA-ECMO or Impella^®^) without the possibility of definitive care using an LV-assist device (LVAD) or heart transplantation.
Multi-Organ Failure	Withdrawal of therapy due to a multi-organ failure (for example in the context of an uncontrollable septic shock) or persistent liver failure.
Respiratory Failure	Withdrawal of care based on respiratory failure. Respiratory failure with hypoxemia, hypercapnia or a combination of these despite maximum support with respirator plus VA-ECMO or even VVA-ECMO.
Presumed Patients Will	This category includes the withdrawal if the patient’s presumed will was against resuscitation. Or, Withdrawal of therapy and termination of intensive care treatment due to an expected poor quality of life (e.g., in context of previously existing serious illness such as dementia or an advanced cancer disease).

VA-ECMO—veno-arterial extracorporeal membrane oxygenation; LVAD—left ventricular assist device; TTM—targeted temperature management; VVA-ECMO—veno-veno-arterial extracorporeal membrane oxygenation.

**Table 2 membranes-11-00270-t002:** Patients’ characteristics at time of admission.

Parameter	(1) Whole Cohort	(2) Survivor	(3) Non-Survivor	p-Value (2) vs. (3)
**Number of Patients**	274 (100%)	71 (25.9%)	203 (74.1%)	
**Mean Age [years]**	60.0 ± 14.3	59.8 ± 14.6	60.0 ± 14.2	0.9085
**Female Gender**	73 (26.6%)	22 (32.0%)	50 (24.6%)	0.2720
**Low-Flow Time [min]**	53.8 ± 28.9	46.0 ± 27.9	56.6 ± 28.8	0.0086
**Shockable Initial Rhythm**	129 (47.1%)	42 (59.2%)	87 (42.8%)	0.0162
**SAPS 2**	70.7 ± 7.4	70.0 ± 8.0	71.0 ± 7.2	0.3187
**SOFA**	11.7 ± 2.3	11.6 ± 2.4	11.8 ± 2.2	0.5266
**SAVE**	−0.47 ± 4.5	0.35 ± 4.8	−0.76 ± 4.4	0.0743
**PREDICT Score 6h [%]**	32 ± 22	42 ± 17	29 ± 23	<0.0001
**Out-of-Hospital Cardiac Arrest**	127 (46.4%)	35 (49.3%)	92 (45.3%)	0.5631
**Initial Lactate Level [mmol/L]**	10.6 ± 7.8	7.9 ± 4.5	11.6 ± 8.4	0.0005
**Comorbidities:**				
** Arterial Hypertension**	104 (38.0%)	30 (42.5%)	74 (36.4%)	0.3860
** Chronic Kidney Disease**	48 (17.5%)	12 (16.9%)	36 (17.7%)	0.8738
** Chronic Liver Disease**	16 (5.8%)	6 (8.5%)	10 (4.9%)	0.2756
** Chronic Lung Disease**	38 (13.9%)	10 (14.1%)	28 (13.8%)	0.9512
** Coronary Artery Disease**	152 (55.5%)	41 (57.8%)	111 (54.7%)	0.6545
** Diabetes Mellitus**	61 (22.3%)	16 (22.5%)	45 (22.2%)	0.9489
** Hypercholesterinemia**	52 (19.0%)	15 (21.1%)	37 (20.3%)	0.5917
** Nicotine**	68 (24.8%)	25 (35.2%)	43 (23.6%)	0.0185
** Peripheral Artery Disease**	17 (6.2%)	7 (9.9%)	10 (4.9%)	0.1380
** Positive Cardiovascular** **Family History**	25 (9.1%)	6 (8.5%)	19 (10.4%)	0.8189

Data given in mean ± standard deviation or number of patients (percentage). p-values are calculated between survivors and non-survivors. (significance level p-value < 0.05).

**Table 3 membranes-11-00270-t003:** Outcome and clinical parameters during ICU stay.

Parameter	(1) Whole Cohort	(2) Survivor	(3) Non-Survivor	p-Value (2) vs. (3)
**Hospital Survival**	71 (25.9%)	71 (100%)	0 (0%)	<0.0001
**Lenght of ICU Stay [days]**	9.0 ± 13.9	19.8 ± 16.5	5.2 ± 10.6	<0.0001
**TTM 33 °C**	208 (75.9%)	46 (64.8%)	152 (74.9%)	0.7531
**NSE-Day 1**	96.1 ± 101.4	68.6 ± 48.3	109.3 ± 115.6	0.0158
**NSE-Day 2**	116.5 ± 98.7	74.5 ± 47.7	147.3 ± 114.2	<0.0001
**NSE-Day 3**	136.9 ± 150.2	73.9 ± 76.2	184.5 ± 1773.8	0.0005
**Respiratory Support [h]**	124.7 ± 140.9	228.4 ± 177.7	88.5 ± 104.0	<0.0001
**Renal Replacement Therapy**	59 (21.3%)	14 (19.7%)	45 (22.2%)	0.6565

Data given in mean ± standard deviation or number of patients (percentage). p-values are calculated between survivors and non-survivors. (significance level p-value < 0.05). Abbreviations: NSE—Neuron Specific Enolase.

**Table 4 membranes-11-00270-t004:** Characteristics of different modes of death in ECPR.

	(1) Survivors	(2) Cerebral Damage	(3) Shock	p-Value 1 vs. 2 vs. 3	p-Value 2 vs. 3
**Patients in Group**	71 (25.9%)	69 (25.2%)	105 (38.3%)		
**Age**	59.8 ± 14.5	54.3 ± 16.5	63.2 ± 11.5	0.0003	<0.0001
**Female Gender**	23 (32.4%)	16 (23.2%)	26 (24.8%)	0.4033	0.8124
**OHCA**	35 (49.3%)	47 (70.1%)	37 (35.6%)	<0.0001	<0.0001
**Shockable Rhythm**	42 (59.2%)	34 (49.3%)	42 (40.0%)	0.0434	0.2275
**First Lactate**	7.9 ± 4.5	11.4 ± 5.4	11.1 ± 4.5	<0.0001	0.6899
**Low-Flow Duration**	46.0 ± 27.7	69.3 ± 29.1	52.3 ± 26.8	<0.0001	0.0001
**NSE D 1**	68.6 ± 48.3	140.7 ± 136.0	91.5 ± 100.9	0.0015	0.0446
**S100 D 1**	0.7 ± 1.9	5.4 ± 4.7	2.6 ± 3.7	<0.0001	0.0160
**SAPS2 Score**	70.0 ± 8.0	69.9 ± 7.9	71.3 ± 6.8	0.3478	0.2010
**SAVE Score**	0.4 ± 4.8	0.4 ± 4.2	−1.3 ± 4.3	0.0111	0.0080
**SOFA Score**	11.6 ± 2.4	11.8 ± 2.1	11.6 ± 2.2	0.9081	0.6955
**PREDICT 6h Score**	42 ± 17	25 ± 17	25 ± 20	<0.0001	0.9528

Table comparing patients with hospital survival to those with the two most dominant causes of death in ECPR death following severe cerebral damage and death by uncontrollable shock. Data given in mean ± standard deviation or number of patients (percentage). p-values are calculated either between all three groups or only comparing patients who died. Abbreviations: OHCA—out of hospital cardiac arrest; SAPS—Simplified Acute Physiology Score; SAVE—Survival after Veno-Arterial ECMO; SOFA—Sepsis-related Organ Failure Assessment score; NSE—Neuron Specific Enolase.

## Data Availability

The datasets of this study are available from the corresponding author on reasonable request.

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
