# Peer review of "Mode of Death after Extracorporeal Cardiopulmonary Resuscitation"

_membranes, 2021, doi:10.3390/membranes11040270_

Round 1
Reviewer 1 Report
First of all, the manuscript is of high interest in topic of high level of combination of conservative and novel device assisted resuscitation with results guiding the next level of therapy.
I suggest to be published with minor revisions, beside I would like to assume that the ECLS and PCAS therapy provided by the author’s team is a hard work and is selected for thoroughly overviewed patients with ongoing CPR.
Questions:
Was there any age limit in the protocol to exclude patients needing CPR from ECLS? I would be also useful to compare the patient population with the survival data of resuscitated elderly population e.g. Kovacs E et al: The role of age in post-cardiac arrest therapy in an elderly patient population. Physiol Int. 2020 Jul 20. doi: 10.1556/2060.2020.00027.
p2, line 80-81:
„excluding patients cannulated for V-A ECMO in the operation theatre and treated at a surgical intensive care unit.”
Why these patients have been excluded from the study? If the authors collect all the resuscitated patients, one has to clarify why to exclude this population…
P2, line93-94.
„to transport patients without ROSC with ongoing mechanical chest compressions to our center, where the patients are screened for ECPR”
Please state the complications during the continuous CC during transport.
p3, Line 109-111
.„ For patients without life-threatening bleeding, anticoagulation 108 was provided by intravenous administration of unfractionated heparin aiming at a partial thromboplastin time of 50-60 sec.”
Please state the ratio of oxygenator thrombosis with this target level.
p5 table 2.:
How do the authors explain that there is no difference between the SAPS, SOFA and SAVE scores at admission?
How these data could be correlated to the the survivor and non—survivor groups.
p6 table 2.
What can be the explanation that nicotine users have a better suvival than non—users?
p6 line 185-186
„Many patients died early after 184 cannulation for V-A ECMO, specifically 95/203 (46.8%) within the first 2 and 133/203 (65.5%) with in first 4 days.”
For the patients died within the first 2 days how may authors determine the cause of death? Was the ECLS procedure excluded?
p7 Line 210-211:
„Those dying from shock were older, had lower low flow durations, and lower NSE values on day 1.”
How do the Authors explain that all those suffered from shock had a controversial shorter low flow durations?
p8 table 3.:
what was the reason to use TTM 33C at only max 75.9 %? the timet o ECLS was much longer than 15 mins.
p9 line 258-261
„We reported earlier that 258 patients with cerebral edema visible in CT-scans perfomed early after ECPR had a very poor suvival [26]. This high incidence of cerebral edema might be explained by the long conventional CPR-durations in ECPR patients [14]. „
These 2 sentences are to be deleted since there is no supporting data in this mauscript on them.
p10 line p273.
„Only a minority die from cerebral damage.”
This conclusional sentence should be proven or be deleted since a 34% of cerebral damage is described in the manuscript.
Few typos should be addressed:
Table 2. Line 177: „P-value” sholud be changed to „p-value”
p7 line 206: „NSE and S100 values of day 1 were significantly higer in case of neurological death”: „higer” is to be changed to „higher”
Reviewer 2 Report
Dear Authors,
thank you for this interesting investigation.
I find the manuscript really interesting.
I just would like you to explain more clearly how patients do not gain rosc and and die precociously. I would suggest you to explore also to confront patients survived after 4 days with the non survivors (including among them also the ones till day 4)
Author Response
I just would like you to explain more clearly how patients do not gain rosc and and die precociously.
Thank you for this valuable question.
Your question focuses on the patient could not be connected to an ECLS.
Unfortunately, we cannot make any statements about these. According to the study design, only patients who were treated in the medical intensive care unit could be included. This therefore includes patients with primarily running ECLS. We cannot say anything about patients who died before they reached our ward.
The study design is now explained in more detail in the METHODS part.
The section “Study setting” reads now: Line 89 ff:
This study includes all patients who were admitted between May 2011 and May 2020 to the medical intensive care unit at our university hospital and received an ECPR. Patients, who received an ECPR in the emergency room and were then transferred to a surgical department, were excluded. Furthermore, patients were excluded for whom the therapy had already been stopped before reaching the intensive care unit.
I would suggest you to explore also to confront patients survived after 4 days with the non survivors (including among them also the ones till day 4)
We would like to thank reviewer 2 for this suggestion.
Indeed, comparing patients dying early to other patients might enable us to identify prognostic markers and potential therapeutic targets – and we will be happy to investigate and report and this important topic. As for this research however, we feel that this question is beyond the scope of this research.
